# Tall Fescue (*Schedonorus arundinaceus* (Schreb.) Dumort.) Turfgrass Cultivars Performance under Reduced N Fertilization

**Marco Schiavon** [1], **Cristina Pornaro** [2,*] **and Stefano Macolino** [2]

1 Environmental Horticulture Department, Fort Lauderdale Research and Education Center, University of Florida, Davie, FL 33314, USA; marcoschiavon@ufl.edu
2 Department of Agronomy, Food, Natural Resources, Animals, and Environment, University of Padova, 35020 Legnaro, Padova, Italy; stefano.macolino@unipd.it
* Correspondence: Cristina.pornaro@unipd.it

**Abstract:** The identification of minimal N requirements for sustaining turfgrass quality and functionality became necessary to reduce N fertilization inputs and avoid potential environmental impacts in the European Union. A two year study was conducted at Padova University in Legnaro, northeastern Italy to investigate the performance of four tall fescue (*Schedonorus arundinaceus* (Schreb.) Dumort.) cultivars ('Lexington', 'Rhambler SRP', 'Rhizing star', and 'Thunderstruck') fertilized twice per year at either 75 or 150 kg N ha$^{-1}$ year$^{-1}$. Turfgrass was evaluated every two weeks for turfgrass visual quality, percent green cover (PGR) as well as dark green color index (DGCI) through digital image analysis and normalized difference vegetation index (NDVI). 'Rhizing star' was the only cultivar that showed poor adaptation to the environment, achieving acceptable turfgrass quality (6.0 or higher) only during June and July 2019. Turf fertilized at 150 kg N ha$^{-1}$ year$^{-1}$ generally showed higher performance than 75 kg N ha$^{-1}$ year$^{-1}$, however, the increase in turfgrass quality was mostly negligible and detected only during the winter months. Results suggest that well adapted tall fescue cultivars could be fertilized at 75 kg N ha$^{-1}$ year$^{-1}$ in Northern Italy.

**Keywords:** N requirement; percent green cover; dark green color index; NDVI

## 1. Introduction

Northern Italy, with its mild winters and hot and dry summers, is considered a transition zone where both cool- and warm-season turfgrass species can be grown successfully [1]. However, long winter dormancy periods, favor the use of cool- over warm-season species in those zones, since most of cool-season turfgrasses are able to maintain color year-round [2,3]. Among cool-season species, tall fescue (*Schedonorus arundinaceus* (Schreb.) Dumort.) is one of the most commonly used grasses in the region thanks to its high tolerance to warm temperatures and drought avoidance due to its ability to use water deep in the soil profile, and its shade and salinity tolerance [4–9]. It can also be successfully used in heavily trafficked turf areas due to its high wear tolerance similar to perennial ryegrass [10].

Reducing N fertilization in agriculture to avoid potential environmental impacts has been encouraged in current European Union Policies [11]. Nitrogen is the most required element for turfgrass growth and development [12]. However, its inconsistent availability in the soil makes seasonal N fertilizations necessary to maintain acceptable turfgrass quality throughout the year. Suggested yearly rate of N fertilization for tall fescue has been reported to be between 200 and 270 kg N ha$^{-1}$ year$^{-1}$ [13,14]. For better turfgrass quality, the suggested N rate should be split in at least two applications, one in the spring and one in the fall [13]. Nevertheless, Grossi et al. [15], investigating timing of fall N fertilization in Italy, concluded that a single application of 100 kg N ha$^{-1}$ through a quick release N source such as ammonium sulfate, was not enough to maintain acceptable tall fescue quality and color through the whole winter. Similar results were observed in Maryland by

Dernoeden et al. [16], who concluded that two applications of 49 kg N ha$^{-1}$ increase tall fescue quality compared to one single application during the winter months, but not during the summer. Pirchio et al. [17], demonstrated that one single application of ammonium sulfate at 150 kg N ha$^{-1}$ increased tall fescue quality, and decreased disease percent as well as weed cover incidence up to 32 weeks after application compared to 75 kg N ha$^{-1}$.

Recently, Schiavon et al. [18], demonstrated that one single application of controlled-release N fertilizers can help sustain bermudagrass (*Cynodon* spp.) quality throughout the growing season. Identifying minimum N rates will help turfgrass managers lower management costs and minimize potential negative effects to the environment [19]. Teuton et al. [20], in a study conducted in Tennessee observed that two varieties of tall fescue, "Dinasty" and "Kentucky 31", responded similarly to different N rates (50, 150, and 300 kg ha$^{-1}$ y$^{-1}$). However, there is still limited information whether different tall fescue cultivars respond equally to low N fertilization rate. The objective of this study was to investigate the long-term performance of four tall fescue cultivars currently commercialized in the local turf market fertilized at either 75 or 150 kg N ha$^{-1}$ year$^{-1}$ using a controlled-release fertilizer to minimize both management and environmental costs.

## 2. Materials and Methods

A field experiment was conducted on mature tall fescue turf at the Experimental Agricultural Farm of Padova University in Legnaro, northeastern Italy (45°20′ N, 11°57′ E; 8 m asl). The soil at the site was a coarse-silty, mixed, mesic, Oxyaquic Eutrudept [21] containing 26.8% clay, 28.5% silt, and 44.7% sand, with a pH of 8.2, 2.68% organic matter, a C/N ratio of 11.9, a total N content of 1.4 mg g$^{-1}$ (combustion method), an Olsen P content of 5.3 mg kg$^{-1}$, and an exchangeable K content of 165.3 mg kg$^{-1}$ (buffered BaCl$_2$ method). The area is described as a humid subtropical climate. Annual minimum, average and maximum temperature are 8.0, 12.6, and 17.4 °C, respectively, and annual average precipitation amount to 831 mm year$^{-1}$ (54 years data) [22]. Monthly precipitation and air temperatures during the investigation period are reported in Table 1.

**Table 1.** Monthly mean air temperatures and monthly precipitations of the study period, and long-term averages (1964–2018) at the agricultural experimental farm of Padova University in Legnaro, northeastern Italy (45°20′ N, 11°57′ E).

| Month | Air Temperature (°C) | | | | Precipitation (mm) | | | |
|---|---|---|---|---|---|---|---|---|
| | 2017 | 2018 | 2019 | 1964–2018 | 2017 | 2018 | 2019 | 1964–2018 |
| January | 0.9 | 5.5 | 2.2 | 2.2 | 14 | 17 | 9 | 52 |
| February | 6.3 | 3.8 | 6.0 | 4.2 | 7 | 64 | 40 | 54 |
| March | 11.1 | 7.1 | 9.8 | 8.1 | 12 | 139 | 10 | 62 |
| April | 13.7 | 15.8 | 13.1 | 12.1 | 56 | 30 | 131 | 68 |
| May | 18.2 | 19.7 | 14.8 | 17.0 | 40 | 70 | 201 | 77 |
| June | 23.7 | 22.9 | 25.0 | 20.6 | 45 | 90 | 9 | 80 |
| July | 24.6 | 24.6 | 24.5 | 22.8 | 45 | 101 | 82 | 74 |
| August | 25.3 | 25.2 | 24.5 | 22.3 | 8 | 109 | 16 | 72 |
| September | 17.8 | 20.7 | 19.6 | 18.3 | 145 | 16 | 68 | 74 |
| October | 13.7 | 15.6 | 15.7 | 13.2 | 7 | 142 | 61 | 80 |
| November | 8.1 | 10.2 | 10.6 | 7.4 | 94 | 63 | 150 | 82 |
| December | 2.8 | 3.0 | 5.7 | 2.9 | 44 | 12 | 90 | 58 |
| Year | 13.9 | 14.5 | 14.3 | 12.6 | 517 | 853 | 867 | 820 |

Four tall fescue cultivars ('Lexington', 'Rhambler SRP', 'Rhizing star', and 'Thunderstruck') were seeded in September 2016 at a rate of 40 g m$^{-2}$. The cultivars were chosen for their widely utilization on the turf Italian market. During establishment turfgrass was fertilized with 50 kg N ha$^{-1}$ and irrigated every other day with 5 mm of water using a sprinkler system.

Once established plots were mowed weekly from March until November at 45 mm using a rotary mower (HRD536, Honda Europe Power Equipment, Atessa-Chieti, Italy)

with clippings removed and fertilized twice (March and May) with urea (46% N) at a rate of 50 kg ha$^{-1}$ for application. Plots were irrigated at 80% reference evapotranspiration (ETo) from June until August every year with weekly irrigation events. From September 2017 plots were fertilized at either 75 or 150 kg N ha$^{-1}$ year$^{-1}$ with a controlled-release N fertilizer (Granucote CRF, (23−5−12, MIVENA)), with high proportion of N in urea form (urea (17.6%), ammonium (2.3%), and nitrate (3.1%)). As reported by the producer, this fertilizer has a longevity of 5−6 months and 77% of the total nitrogen is coated. Turf received fertilization twice per year in September and March. Grass weeds were manually removed, while broadleaf weeds were controlled using Vithal Turfene plus selective herbicide (Dicamba/MCPP-P). Disease control strategy did not include pesticide applications.

From September 2017 until September 2019, plots were evaluated every 2 weeks for turfgrass visual quality on a scale from 1 = worst to 9 = best, with 6 = minimally acceptable quality [23], percent green cover (PGR) as well as dark green color index (DGCI) through digital image analysis (DIA) [24], and normalized difference vegetation index (NDVI; FieldScout CM 1000; Spectrum Inc., Aurora, IL, USA). Digital images were made using a Canon Powershot G12 installed on a light box. As reported by Richardson et al. [24] camera settings included a shutter speed of 1/400 s, an aperture of F4.0, and a focal length of 32 mm, while the indicated hue range was from 57 to 107 and the saturation range from 0 to 100. To allow comparison between months of different years, biweekly data were subsequently averaged over month. Data from September 2017 to September 2018, and from September 2018 to September 2019 represented the first (Year 1) and the second (Year 2) year of experimentation respectively. Plots were arranged in a split-plot design with three replications. Cultivars served as main plot (2 by 3 m) and N rate served as split plot (1 by 3 m). Analysis of variance (ANOVA) were performed for each detected parameter with a repeated measure analysis using a compound symmetry covariance structure (Statistical Analysis Software, SAS Proc Mixed version 9.4; SAS Institute, Cary, NC, USA). Normality and homoscedasticity of residuals were checked by using graphical analyses. When appropriate, means were separated with Fisher's protected least significant difference at 0.05 probability level.

## 3. Results

The interaction of year, month, and cultivar, and the interaction of year and N rate affected turfgrass quality (Table 2). 'Thunderstruck' was the cultivar with the highest visual quality during the first three months of the trial, followed by 'Lexington' and 'Rhambler SRP'. No differences among these three cultivars were detected during the first year of the trial. Conversely, 'Rhizing star' was the cultivar with the lowest visual quality and never reached acceptable quality during the first year of the study. 'Thunderstruck' also showed highest visual quality during fall 2018 and spring 2019. Visual quality of 'Thunderstruck', 'Lexington', and 'Rhambler SRP' dropped below acceptable levels only during winter. Lowest quality ratings were recorded in 'Rhizing star' from September 2018 until January 2019, and in July and August 2019 (Figure 1). However, although the higher N rate translated into better turfgrass quality only in year 2 (Figure 2). The lower N rate showed lower turfgrass quality compared to 150 kg N ha$^{-1}$ during the second year of the trial, it still maintained sufficient turfgrass quality (Figure 2). Conversely, no differences were detected between the two N rates during the first year, when turfgrass quality did not achieve sufficient ratings, regardless of the N rate.

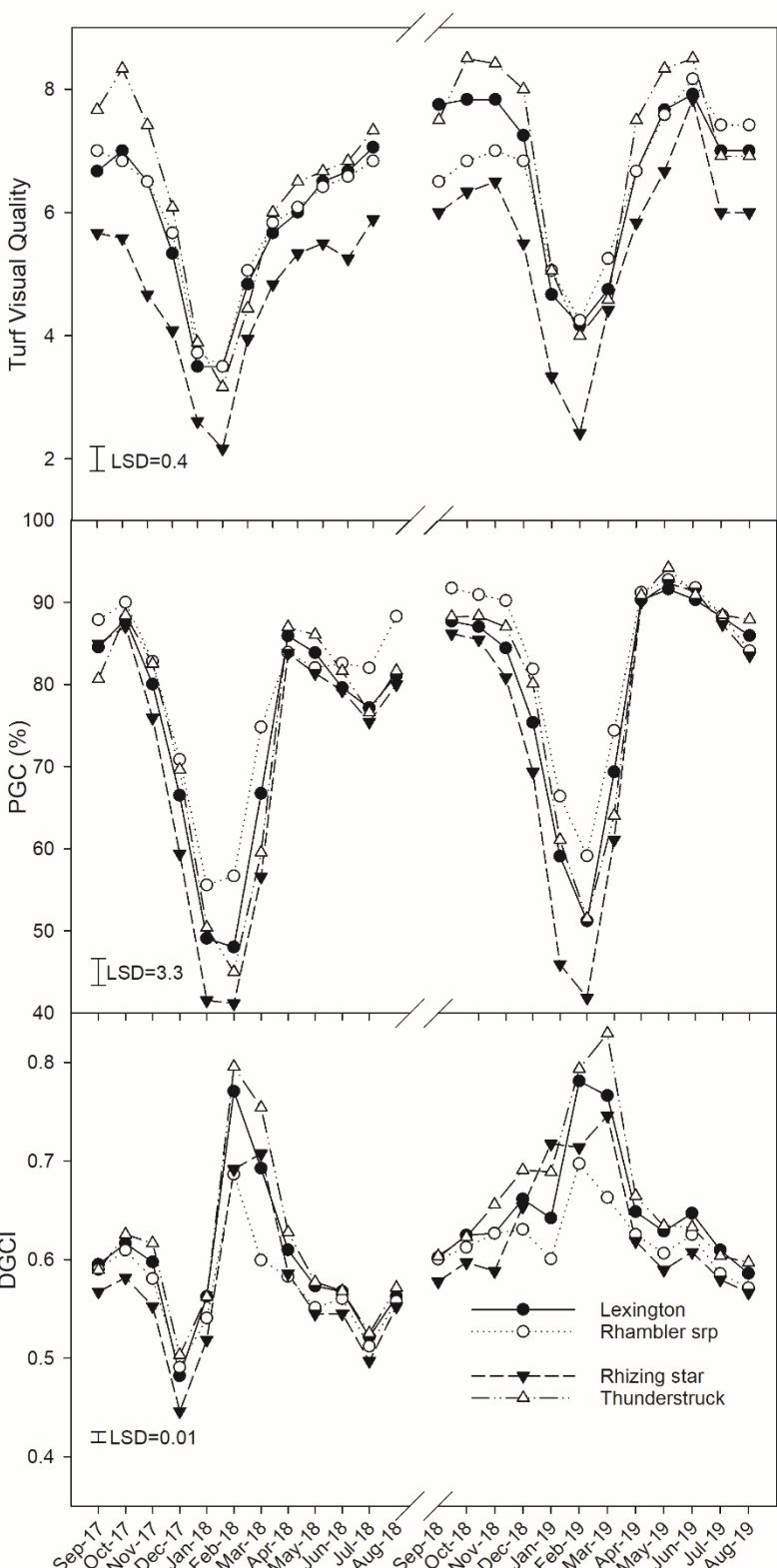

**Figure 1.** Turfgrass quality, percent green cover (PGC), and dark green color index (DGCI) of four tall fescue cultivars ('Lexington', 'Rhambler SRP', 'Rhizing star', and 'Thunderstruck') as affected by year and month. Data are averaged over 2 N rates and 3 replicates, and represent an average of 6 data points. Turfgrass quality was visually assessed (1−9 scale), PGC and DGCI were determined through digital image analysis.

**Table 2.** Results of analysis of variance testing the effects of cultivar, N rate, year, and month and their interactions on turfgrass quality, percent green cover (PGC), dark green color index (DGCI), and normalized difference vegetation index (NDVI) of tall fescue [‡].

|  | Turfgras Quality [‡] | PGC | DGCI | NDVI |
|---|---|---|---|---|
| Cultivar (C) | NS [†] | * | NS | NS |
| N rate (N) | * | *** | NS | * |
| C × N | NS | NS | NS | NS |
| Year (Y) | *** | *** | *** | NS |
| Y × C | *** | * | *** | ** |
| Y × N | ** | NS | NS | NS |
| Y × C × N | NS | NS | NS | NS |
| Month (M) | *** | *** | *** | *** |
| M × C | *** | *** | NS | NS |
| M × N | NS | *** | *** | *** |
| M × C × N | NS | NS | NS | NS |

* Significant F test at the 0.05 level of probability. ** Significant F test at the 0.01 level of probability. *** Significant F test at the 0.001 level of probability. [†] NS, nonsignificant at the 0.05 probability level. [‡] Turfgrass quality was visually assessed (1−9 scale), PGC and DGCI were determined through digital image analysis, and NDVI was measured with Field-Scout CM 1000 NDVI meter.

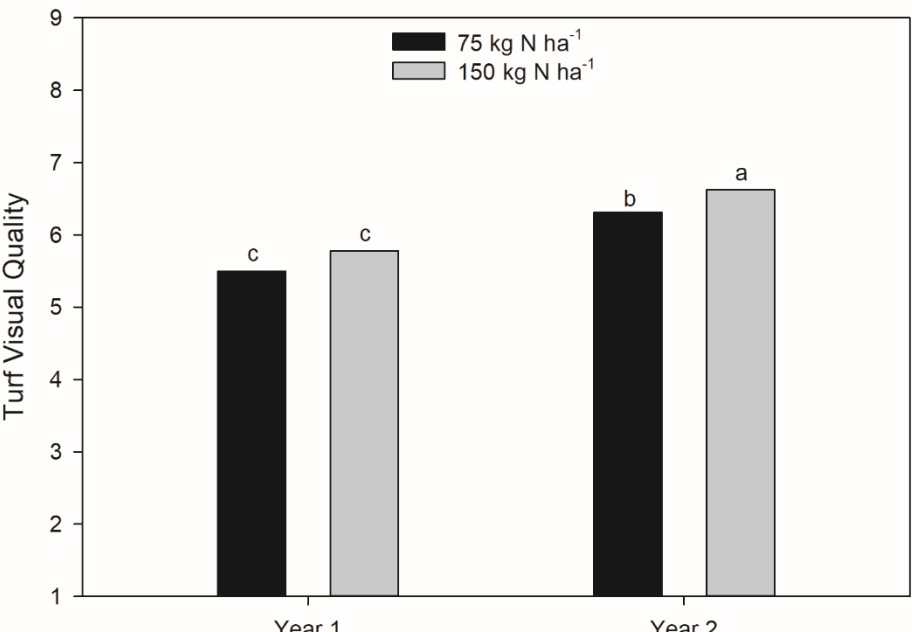

**Figure 2.** Turfgrass quality as affected by year. Data are averaged over 2 N rates, 12 months, and 3 replicates, and represent an average of 72 data points. Turfgrass quality was visually assessed (1−9 scale). Mean values with the same letters are not significantly different from one other (Fisher's protected LSD test at the 0.05 probability level).

Percent green cover was affected by the interaction of year, month and cultivar, and by the interaction of month and N rate. When data were averaged over N rate, PGC rate mirrored that of visual quality ratings, with lowest PGC recorded during the winter months. 'Rhambler SRP' was the cultivar that showed higher PGC, specially during the winter months (Figure 1). Conversely, 'Rhizing star' was the cultivar with the lowest PGC in January and May (Figure 1). However, no cultivar went fully dormant during winter, and all cultivars covered at least 40% of the ground even in February (Figure 1). In fact, during January and February minimum temperatures were lower than 0 only twice in 2018 (with minimum temperature of −1 °C for no more than 4 days) and once in 2019 (with minimum temperature of −1 °C for no more than 3 days). When data are presented for each combination of month and N rate, PGC collected from November until July were

higher in tall fescue fertilized at 150 kg N ha$^{-1}$ compared to that fertilized at 75 kg N ha$^{-1}$ (Figure 3). However, the widest gap between N fertilization treatments was observed from December until March, when plots fertilized at 150 kg N ha$^{-1}$ covered 5% more ground than plots fertilized at 75 kg N ha$^{-1}$.

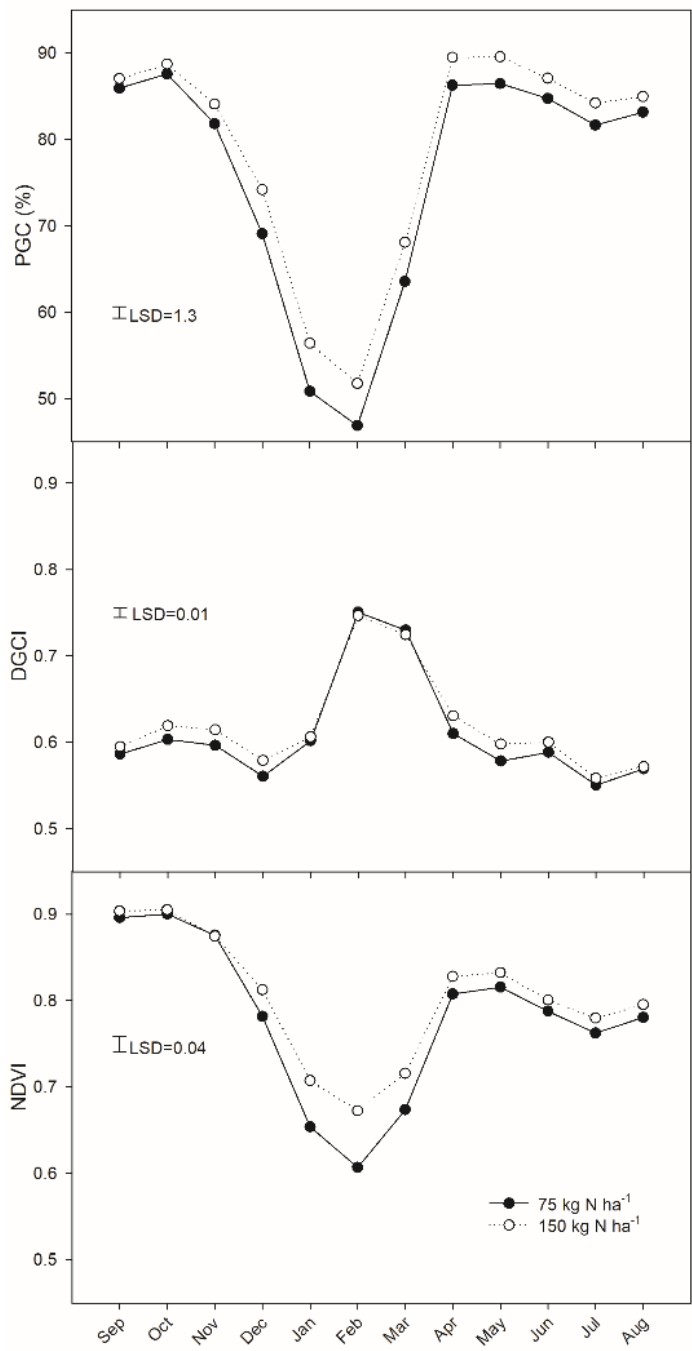

**Figure 3.** Percent green cover (PGC), dark green color index (DGCI), and NDVI of four tall fescue cultivars ('Lexington', 'Rhambler SRP', 'Rhizing star', and 'Thunderstruck') as affected by month. Data are averaged over 2 N rates, 2 years, and 3 replicates, and represent an average of 12 data points. Percent green cover and DGCI, were determined through digital image analysis, NDVI was measured with Field-Scout CM 1000 m.

Dark green color index corroborated turfgrass visual quality (Figure 1). 'Rhizing Star' showed lower DGCI at the beginning of the study and during February and March of both years. 'Rhambler SRP' had the highest DGCI in February 2018, while the highest DCGI was

detected in 'Thunderstruck' in February 2019 (Figure 1). When data presented separately for month and N rate, DGCI increased during the winter month (Figure 3). The higher N rate showed higher DGCI only during March; no differences were detected between N rates during the rest of the year (Figure 3).

The interaction of month and N rate, and the interaction of year and cultivar affected NDVI (Table 3). Unlike DGCI, NDVI was higher in tall fescue fertilized at 150 kg N ha$^{-1}$ compared to the 75 kg N ha$^{-1}$ rate from December until March. No differences were detected between the two N rates from April until November (Figure 3). When data were presented separately for each combination of year and cultivar, 'Rhizing star' showed the lowest NDVI during the first year of the trial, while no differences were detected among 'Lexington', 'Rhambler SRP', and 'Thunderstruck'. Similar results were found during the second year of the study; however, no differences were detectable between 'Lexington' and 'Rhizing star' during year two (Table 3).

**Table 3.** NDVI [‡] of four tall fescue cultivars ('Lexington', 'Rhambler SRP', 'Rhizing star', and 'Thunderstruck') during the two year of the trial. Data are averaged over 2 N rates, 12 months and 3 replicates, and represent an average of 72 data points.

| | NDVI [‡] | |
|---|---|---|
| | **Year 1** | **Year 2** |
| Lexington | 0.80 a [†] | 0.78 ab |
| Rhambler SRP | 0.83 a | 0.79 a |
| Rhizing star | 0.76 b | 0.76 b |
| Thunderstruck | 0.80 a | 0.79 a |

[†] Valued followed by the same letter in a column are not statistically different α = 0.05; [‡] NDVI was measured using FieldScout CM 1000 m.

## 4. Discussion

The need to reduce N fertilization to prevent negative impacts on the environment, coupled with prioritizing resources from landscape areas to crops that can provide food for human consumption, increases the need for reducing inputs in urban landscapes [25–27]. Moreover, despite the importance of N fertilization for turfgrass growth and maintenance, increasing N fertilizers costs and potential negative environmental implications [19] limit the amount of N that can be applied to turfgrass areas. Identifying the cultivars that are well adapted to the environment where they are being grown can help sustain turfgrass aesthetic and functionality while minimizing input costs.

Tall fescue cultivars in this study seem to have the same N requirements, since cultivar effect never interacted with N rate (Table 2). Nevertheless, tall fescue fertilized at 150 kg N ha$^{-1}$ performed slightly better than at 75 kg N ha$^{-1}$ (Figures 2 and 3). However, the increase in turfgrass quality was mostly negligible, and positive effects of the higher fertilization rate were visible only during the winter (Figure 3). The economic and environmental impact that higher N application could have in the face of such a small increase in turfgrass quality cannot be justified by stakeholders.

In this study differences among cultivars were often visible (Figure 1); 'Thunderstruck' seemed to possess the best adaptation to the northern Italian climate, followed by 'Lexington' and 'Rhambler SRP'. Conversely, 'Rhizing star' achieved acceptable ratings only during the second year of the trial. Similar performance of new turf-type tall fescue cultivars has been previously documented in Italy [25,28], suggesting great adaptability of this species in the country. Dark green color index corroborated visual quality ratings (Figure 1). The increase of DGCI during the winter months concurs with the decrease of PGC (Figure 1). This event is possibly due to the different physiological status of the plant during winter. Rorie et al., [29] demonstrated that DGCI highly correlates to plant N status. Hence, higher DGCI during the winter months are possibly due to high N content in leaf tissues since the plant is not actively growing, but not totally dormant. More research is

needed to identify if these cultivars have similar tolerance to environmental stresses that may arise in the area such as frost or drought.

## 5. Conclusions

Tall fescue is the most used turfgrass species in the European transition zone, therefore it is a high priority to identify appropriate management practices to maintain turfgrass quality and functionality while reducing potential negative consequences on the environment. The present study revealed that no great differences are detectable among the studied turf-type tall fescue cultivars in northern Italy. Moreover, urea-based controlled-release N fertilizer applied twice a year provides tall fescue with sufficient quality from spring until the fall. Our results also suggest that a total of 75 kg N ha$^{-1}$ year$^{-1}$ could be sufficient to maintain tall fescue in the transition zone of southern Europe.

**Author Contributions:** Conceptualization, S.M.; methodology, S.M.; software, M.S.; validation, S.M. and M.S.; formal analysis, M.S.; investigation, C.P.; resources, S.M.; data curation, M.S.; writing—original draft preparation, M.S.; writing—review and editing, S.M., C.P., and M.S.; visualization, M.S.; supervision, S.M.; project administration, S.M.; funding acquisition, S.M. All authors have read and agreed to the published version of the manuscript.

**Funding:** This research was funded by the University of Padova (DOR1889925/18) "Comportamento di cultivar da tappeto erboso *di Festuca arundinacea* in condizioni di limitata concimazione azotata".

**Institutional Review Board Statement:** Not applicable.

**Informed Consent Statement:** Not applicable.

**Acknowledgments:** The Authors thank Padana Sementi Elette s.r.l. for providing the seed, the staff of the Experimental Agricultural Farm of the University of Padova for the efficient work in managing field plots, and Nicola Faraon for help in the field activity.

**Conflicts of Interest:** The authors declare no conflict of interest.

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
