# Peer review of "Tall Fescue (Schedonorus arundinaceus (Schreb.) Dumort.) Turfgrass Cultivars Performance under Reduced N Fertilization"

_agronomy, doi:10.3390/agronomy11020193_

Round 1

Reviewer 1 Report

I attached the manuscript with my comments and highlighted parts which need some improvements, explanation or supplementation. There are several misprints and formal shortcomings (superscripts and subscripts). Some important information which should be mentioned in methods are shown in following chapters (e.g. number of repetitions). In Table A, there are presented meteodata as an average of 54 years while in the text above there is a reference that data are from 25 years. 

As the manuscript is aimed to comparison of tall fescue cultivars, I miss the explanation why just these 4 cultivars were selected and if there are some substantial differences among them (as there are hundreds of TF cultivars registered). 

Authors mentioned that no cultivar went to dormancy during winter, but for generalisation, reader would need to now extreme low temperatures during the tested winter periods.

Author Response

Reviewer: 1

I attached the manuscript with my comments and highlighted parts which need some improvements, explanation or supplementation. There are several misprints and formal shortcomings (superscripts and subscripts). Some important information which should be mentioned in methods are shown in following chapters (e.g. number of repetitions). In Table A, there are presented meteodata as an average of 54 years while in the text above there is a reference that data are from 25 years.

As the manuscript is aimed to comparison of tall fescue cultivars, I miss the explanation why just these 4 cultivars were selected and if there are some substantial differences among them (as there are hundreds of TF cultivars registered).

Authors mentioned that no cultivar went to dormancy during winter, but for generalisation, reader would need to now extreme low temperatures during the tested winter periods..

For the response see specific comments:

Line 29: changed accordingly.

Line 50: It is an interesting topic of discussion, but it seems out of context, so we prefer not to deepen it in the manuscript. However, for clarity, we have specified in the objective of the study that we aimed to reduce both environmental and management costs.

 Line 55: corrected.

Line 60: clarified.

Line 67: changed accordingly.

Lines 68 and 71: corrected.

Lines 70-71: data has been changed with 54-yr averages.

Line 79: fertilization information has been better described.

Line 99: number of replications are reported in line 109.

Line 100: corrected.

Line 144: information on minimum temperatures have been added in lines 161-164.

Line 178: Changed to: ‘The need to reduce N fertilization to prevent negative impacts on the environment, coupled with prioritizing resources from landscape areas to crops that can provide food for human consumption, increasing the need for reducing inputs in urban landscapes [25-27]’

Line 184: the reason for cultivars choice has been added at lines 81-82.

Line 190: changed to: ‘cannot be justified by stakeholders’

Line 209 a: the text has been changed with “among the studied turf-type tall fescue cultivars”.

Line 209 b: changed accordingly.

Reviewer 2 Report

Overall I thought this was an interesting article that was well written and easy to understand. Thank you for doing this work.  There are a few questions I have after reading this article that I would like to see addressed before this should be published as well as a couple of grammatical edits.

Big picture:

I would like to see a greater description of the fertilizer used in the experiment. What is the release curve of this fertilizer? With only two applications per year was it a 120 day release curve? Not being from Italy, I wasn't familiar with the product used to know if that played a role in the performance. Was there a coating or more of an organic product that made it slow? In my research I see there is a portion that is not coated, you should mention this as well. Did you take any tissue samples? That could have helped out if the lower rate had less nitrogen in the plant or similar amounts and would make it easier to see the higher rate was not necessary. 

Specific line edits:

Line 21: I think that should be months and not month.

Line 26: Change turf to turfgrass since this is the first mention in the actual article.

Line 31: Add es after grass.

Line 36: What is UE? Define for those not in Europe.

Line 44: Should this be sulfate and not sulphate?

Line 49: Please clarify: and decreased disease percent weed cover.

Line 55: Change varsities to varieties.

Line 71: The -1 after year should be superscript.

Table 1. Consider adding the coordinates after the location for people to easily find on a map.

Line 79: Add an a after the word using.

Please add more detail on the digital image. Did you use a light box? This is critical if you did DGCI. Also what were your thresholds for hue and saturation?

Line 118: Consider changing: it still harvested sufficient to: it still maintained sufficient.

Table 2. Please add footnotes about how each of the variables were determined so that these tables can stand alone. The same request for all other tables and figures.

In figure 2 please add Visual out front.

Author Response

Reviewer: 2

Overall I thought this was an interesting article that was well written and easy to understand. Thank you for doing this work.  There are a few questions I have after reading this article that I would like to see addressed before this should be published as well as a couple of grammatical edits.

Big picture:

I would like to see a greater description of the fertilizer used in the experiment. What is the release curve of this fertilizer? With only two applications per year was it a 120 day release curve? Not being from Italy, I wasn't familiar with the product used to know if that played a role in the performance. Was there a coating or more of an organic product that made it slow? In my research I see there is a portion that is not coated, you should mention this as well. Did you take any tissue samples? That could have helped out if the lower rate had less nitrogen in the plant or similar amounts and would make it easier to see the higher rate was not necessary.

More information on fertilizer has been added at lines 91-92. Information came from producer brochure. We did not take tissue samples on the fertilizer.

Specific line edits:

Line 21: I think that should be months and not month.

Changed.

Line 26: Change turf to turfgrass since this is the first mention in the actual article.

Changed.

Line 31: Add es after grass.

Changed.

Line 36: What is UE? Define for those not in Europe.

Changed.

Line 44: Should this be sulfate and not sulphate?

Changed.

Line 49: Please clarify: and decreased disease percent weed cover.

The sentence has been improved.

Line 55: Change varsities to varieties.

Changed.

Line 71: The -1 after year should be superscript.

Changed.

Table 1. Consider adding the coordinates after the location for people to easily find on a map.

Added.

Line 79: Add an a after the word using.

Added.

Please add more detail on the digital image. Did you use a light box? This is critical if you did DGCI. Also what were your thresholds for hue and saturation?

More details have been added.

Line 118: Consider changing: it still harvested sufficient to: it still maintained sufficient.

Changed.

Table 2. Please add footnotes about how each of the variables were determined so that these tables can stand alone. The same request for all other tables and figures.

Information have been added.

In figure 2 please add Visual out front.

Changed

Round 2

Reviewer 1 Report

I appreciate the changes which were made and improved the readibility of the manuscript. I have found just one misprint in the text (ANOVA - page 3, line 110), but there is a statement I cannot agree with. On the page 8, line 201, there is still the claim that resources (N fertilizers in this case) are preferentialy use on agricultural land instead of turfgrasses. I did not mentioned it yet (due to their scarcity). 

Author Response

Dear Reviewr,

Thank you for your time in reviweing our manuscript. We have changed "ANVOA" with "ANOVA" at line 110.

About your comment at line 201, the sentence has been modified and it does not mention resource scarcity. All over the world agricultural use of resources is prioritized over landscape areas.